# *Lonicera caerulea* Extract Attenuates Non-Alcoholic Fatty Liver Disease in Free Fatty Acid-Induced HepG2 Hepatocytes and in High Fat Diet-Fed Mice

**DOI:** 10.3390/nu11030494

**Published:** 2019-02-26

**Authors:** Miey Park, Jeong-Hyun Yoo, You-Suk Lee, Hae-Jeung Lee

**Affiliations:** 1Institute for Aging and Clinical Nutrition Research, Gachon University, Gyeonggi-do 13120, Korea; mieyp@naver.com (M.P.); yjh8252@naver.com (J.-H.Y.); ysleeyun@gachon.ac.kr (Y.-S.L.); 2Department of Food and Nutrition, Gachon University, Gyeonggi-do 13120, Korea

**Keywords:** Honeyberry, free fatty acids, nonalcoholic fatty liver disease, HepG2 cells, high fat diet

## Abstract

Honeyberry (*Lonicera caerulea*) has been used for medicinal purposes for thousands of years. Its predominant anthocyanin, cyanidin-3-O-glucoside (C3G), possesses antioxidant and many other potent biological activities. We aimed to investigate the effects of honeyberry extract (HBE) supplementation on HepG2 cellular steatosis induced by free fatty acids (FFA) and in diet-induced obese mice. HepG2 cells were incubated with 1 mM FFA to induce lipid accumulation with or without HBE. Obesity in mice was induced by a 45% high fat diet (HFD) for 6 weeks and subsequent supplementation of 0.5% HBE (LH) and 1% HBE (MH) for 6 weeks. HBE suppressed fatty acid synthesis and ameliorated lipid accumulation in HepG2 cells induced by FFA. Moreover, HBE also decreased lipid accumulation in the liver in the supplemented HBE group (LH, 0.5% or MH, 1%) compared with the control group. The expressions of adipogenic genes involved in hepatic lipid metabolism of sterol regulatory element-binding protein-1 (SREBP-1c), CCAAT/enhancer-binding protein alpha (C/EBPα), peroxisome proliferator-activated receptor gamma (PPARγ), and fatty acid synthase (FAS) were decreased both in the HepG2 cells and in the livers of HBE-supplemented mice. In addition, HBE increased mRNA and protein levels of carnitine palmitoyltransferase (CPT-1) and peroxisome proliferator-activated receptor α (PPARα), which are involved in fatty acid oxidation. Furthermore, HBE treatment increased the phosphorylation of AMP-activated protein kinase (AMPK) and Acetyl-CoA Carboxylase (ACC). Honeyberry effectively reduced triglyceride accumulation through down-regulation of hepatic lipid metabolic gene expression and up-regulation of the activation of AMPK and ACC signaling in both the HepG2 cells as well as in livers of diet-induced obese mice. These results suggest that HBE may actively ameliorate non-alcoholic fatty liver disease.

## 1. Introduction

Non-alcoholic fatty liver disease (NAFLD) is one of the most common liver diseases, resulting from excessive accumulation of lipids in the liver despite a low level of alcohol consumption [1,2]. Recent increases in the prevalence of obesity and obesity-related metabolic disorders have demonstrated parallelism with a global increase in NAFLD [3,4]. More than 20% of adults in western countries are diagnosed with NAFLD, and its prevalence increases from 70 to 90% among people who are obese or suffer from diabetes [5]. Furthermore, NAFLD is significantly associated with an increased risk of cardiovascular diseases [6].

The criterion for diagnosis of NAFLD is the presence of fat in more than 5% of the hepatocytes [7]. Excessive accumulation of triglycerides (TG) due to esterification of glycerol and free fatty acid (FFA) induces the hallmark feature of NAFLD [8]. Once in the hepatocytes, FFA undergo acyl-CoA synthase activity and form fatty acyl-CoAs, which may enter either the esterification or β-oxidation pathways [7]. NAFLD is characterized by the development of oxidative stress and changes in redox balance. Besides alterations in lipid metabolism, it has been speculated that mitochondrial dysfunction, inflammation, and oxidative stress may also be closely associated with the progression of NAFLD [9,10].

The honeyberry (*Lonicera caerulea*, HB) plant originates in high mountains or wet areas and is used as a traditional medicine in northern Russia, China, and Japan [11]. Its fruits are purple-colored, hard berries, which are about 1–2 cm long and 1 cm wide and can resist temperatures below −40 °C [12]. Recently, honeyberry (HB) has been widely harvested in many countries including South Korea and consumed as a part of the human diet; regular intake of HB is associated with health benefits that prevent chronic diseases, such as diabetes mellitus and cardiovascular diseases [13,14]. Recently, numerous studies have indicated that berries rich in polyphenols, such as phenolic acids, anthocyanins, and tannins [12,13,15], can provide advantages in mitigating and preventing metabolic syndrome components including obesity and lipid disorders, thus protecting against NAFLD-related metabolic alterations [16,17].

Cyanidin 3-O-glucoside (C3G), which is the most widespread anthocyanin in nature, is the predominant phenolic compound present in HB [18]. We have previously demonstrated that C3G isolated from HB enhances insulin production and activates insulin signaling [18] and is responsible for hepatoprotective effects through the inhibition of ROS and activation of antioxidant mechanisms [19].

Major cellular sensors for maintaining energy status are AMP-activated protein kinase (AMPK) and sterol regulatory element-binding protein (SREBP). They are critical regulators of hepatic lipid metabolism including glucose transport, gluconeogenesis, and lipolysis [20,21]. Anthocyanins can improve metabolism through the activation of AMP-activated protein kinase (AMPK) [22]. Recently, Kim et al. showed that AMPK inactivates SREBP-1c-dependent hepatic steatosis in high-fat diet (HFD)-induced animal models [23]. Additionally, AMPK promotes fatty-acid oxidation by directly suppressing Acetyl-CoA Carboxylase 1 (ACC1) activity and interfering with its transcriptional activity on fatty acid synthase (FAS) [24]. Peroxisome proliferator-activated receptor α (PPARα), which is the major lipid oxidation regulator in the liver, is stimulated by anthocyanins [25] and reduces hepatic lipid concentrations [26]. In the liver, activated AMPK regulates fatty-acid oxidation by directly suppressing ACC activity and up-regulating the gene expression of carnitine palmitoyltransferase (CPT-1), a key regulator of fatty-acid oxidation [27]. Based on these findings, the present study was designed to provide basic data for a better understanding of the pharmacological effects of HB extract (HBE) on the improvement of NAFLD using FFA-treated HepG2 cells and high fat diet (HFD) obese mouse.

## 2. Materials and Methods

### 2.1. Preparation of Honeyberry Extract (HBE)

Honeyberry (*Lonicera caerulea*, HB) was purchased from the Jeongseon County (Gangwon-do, Korea) at the time of harvest in early June. HB was thoroughly ground with a blender (FM-681C; Hanil Electric, Seoul, Korea) and extracted with 900 mL of 25% ethanol and distilled water at 60 °C, placed in a water bath (C-WBE, Chang Shin Scientific Co. Seoul, Korea) for 3 h. The mixture was filtered through No. 2 filter paper (Toyo Roshi Kaisha, Tokyo, Japan) and was concentrated under reduced pressure by a rotary evaporator (EYELA, N-N series, Tokyo, Japan) at 60–65 °C. The prepared samples were stored at −20 °C until further use [28,29].

### 2.2. Analysis of HBE by HPLC

High-performance liquid chromatography with diode array detection (HPLC-DAD) analysis was conducted on Agilent 1100 DAD system. A C18 Kinetex column (4.5 mm × 25 cm, 5 μm) was used for the separation. The solvent flow rate and injection volume were kept as 1.2 mL/min and 10 µL, respectively, and the column temperature was fixed at 30 °C. The sample was eluted using mobile phase A (5% formic acid and water) and mobile phase B (5% formic acid and Acetonitrile) and was detected at 517 nm. The C3G, a major anthocyanin present in HBE, was detected by HPLC–DAD. The concentration of C3G in HBE was 5.91 mg/g with a 70% yield in a 25% ethanolic extract of honeyberry (Figure 1).

### 2.3. Cell Culture and Induction of Steatosis

HepG2 cell were obtained from American Type Culture Collection (ATCC) and maintained in Dulbecco’s modified Eagle’s medium (DMEM) containing 1% antibiotics (antibiotic–antimycotic, ThermoFisher, USA) and 10% fetal bovine serum in a humidified atmosphere of 5% CO_2_ at 37 °C. The FFA (oleic acid and palmitic acid, 2:1) were dissolved in isopropyl alcohol at 1mM concentrations. After reaching 80% confluence, the HepG2 cells were cultured with serum-free medium containing 1% fat-free bovine serum albumin (BSA) and exposed to 1mM of FFA with BSA. We prepared an HBE stock solution (20 mg/ml) and used 250, 500, and 1000 μg/ml in DMEM medium. HBE was added to the FFA–BSA complex and incubated for 24 h. Control cells were treated with 1% BSA only [30].

### 2.4. Effects of FFA on Cell Cytotoxicity

Effects of HBE and FFA on cell proliferation were determined using Cell Counting Kit-8 (CCK-8) (Dojindo Molecular Technologies, Rockville, MD, USA) [31]. HepG2 cells were seeded in a 96-well plate at a concentration of 5 × 10^4^ cells per well, and then allowed to adhere to the wells. The medium was replaced with serum-free DMEM containing 1mM FFA and 1% BSA, with and without HBE, and incubated further for 24 h. Subsequently, the cells were incubated with 10 μL of CCK-8 and serum-free DMEM at 37 °C for 2 h. The experiment was performed in triplicate. The results were expressed as percentage of viable cells with respect to untreated control cells. The absorbance was measured at 450 nm using a microplate spectrophotometer system (BioRad, Hercules, CA, USA) [32].

### 2.5. Oil Red O Staining

To detect accumulation of intracellular neutral lipids, the control and HBE-treated HepG2 cells were fixed with 10% formalin for 1 h, and then stained with Oil Red O working solution for 30 min at room temperature. The cells were washed three times with deionized water and observed under an inverted microscope. For lipid quantification in HepG2 cells, 100% isopropanol was added to dissolve the Oil Red O reagent and the absorbance was measured at 500 nm. Further, HepG2 cells were incubated in 1mM FFA to induce hepatic steatosis for 24 h, with and without HBE supplementation. The cells were stained with Oil Red O solution and intracellular lipid accumulation was visually observed under a microscope (200×). Control cells were treated with 1% BSA only.

### 2.6. Triglyceride Colorimetric Assay (TG Assay)

HepG2 cells were plated in 6-well plates at 1 × 10^5^ cells per well and then allowed to adhere to wells for 24 h. Later, serum-free DMEM containing 1mM FFA and 1% BSA, with and without HBE supplementation, was added to the wells and plates incubated for 24 h. The TG levels of the cells were measured by TG assay kit (Asanpharm, Seoul, Korea).

### 2.7. Quantification of Gene Expression Using Real-time PCR

Total RNA was isolated from HepG2 and liver tissue using the Total RNA Mini Kit (TaKaRa, Seoul, Korea), according to the manufacturer’s instructions. One microgram of total RNA was used to synthesize cDNA. Real-time PCR was performed using an SYBR® Select Master Mix (Thermo Fisher Scientific, San Jose, CA, USA) using a QuantStudio3 Real-Time PCR system (Thermo Fisher Scientific, San Jose, CA, USA). Primer sequences are shown in Table 1.

### 2.8. Western Blot Analysis for HepG2 Cells

To examine the effects of HBE on FFA-induced steatosis, we observed the expression of SREBP-1c, C/EBPα, PPARγ, and FAS. In addition, we examined AMPK and ACC phosphorylation, which indicates AMPK activation, by Western blot analysis. HepG2 cell lysates (from 1 × 10^6^ cells per well) were extracted by protein lysis buffer (iNtRON Biotechnology, Seongnam-si, Korea) supplemented with protease and phosphate inhibitors. Equal amounts of protein (30 μg) were separated on 10% polyacrylamide gels and subjected to western blot analysis using polyclonal antibodies. PVDF membranes were immunoblotted with primary antibodies: C/EBPα (1:500), PPARγ (1:1000), SREBP-1c (1:1000), FAS (1:100), PPARα (1:1000), ACC (1: 2000), pACC (1: 10000), pAMPK (1:1000), AMPK (1:1000), and CPT1 (1:1000) at 4 °C overnight. After three washes, the membranes were incubated with anti-mouse or anti-rabbit horseradish peroxidase (HRP)-coupled secondary antibodies (1:2500) for 1 h at room temperature in 5% skim milk. All the band-density quantifications were normalized with β-actin. Blots were developed using ECL western blot detection kit (Amersham Pharmacia, Little Chalfont, Buckinghamshire, UK) and visualized using the ImageQuantTM LAS500 system (Amersham Pharmacia, Little Chalfont, Buckinghamshire, UK).

### 2.9. Animal Diets and Preparations of Tissue Sample

Five-week-old imprinting control region (ICR) male mice were obtained from Orient Bio Co. (Seongnam-si, Korea). The scheme of animal experimental protocol showed in Appendix A. After a one-week acclimatization period, the mice were randomly divided into two groups (control group with normal diet (AIN-93G, D10012G, Research diet, New Brunswick, NJ, USA) and obesity-induced group fed for 6 weeks with 45% high fat diet (HFD; D12451, Research diet, New Brunswick, NJ, USA)). They had free access to water and the diet throughout the study. After the obesity-induction period, mice (*n* = 7 per group) were re-randomized into four groups: normal diet control group (ND), high fat diet control group (HC), HFD group fed with 0.5% HBE (LH, low dose of HBE), and HFD group fed with 1% HBE (MH, middle dose of HBE) for 6 weeks. Diet composition of HFD and HBE treatment groups was shown in Appendix A. Food intake was calculated by subtracting the remaining amount of food from the supplied amount of food. Food intake was measured every two days at 10 to 11 a.m. and body mass was measured weekly for 6 weeks. All experiments were performed with the approval from Gachon University using guidelines for the care and use of laboratory animals (reference number: R2017007). After overnight fasting, final body weights were monitored, and all mice were euthanized using CO_2_. Blood was collected by cardiac puncture and placed into blood collecting tubes. To collect serum, blood was separated by centrifugation at 3000 rpm for 15 min at 4 °C (Combi-514R, Hanil Co. Ltd., Seoul, Korea) and stored at −80 °C. Liver tissues were removed, rinsed with physiological saline (PBS), and stored immediately at −80 °C for future use. The liver tissues were fixed in 10% neutral buffered formalin (Sigma-Aldrich, St. Louis, MO, USA) for paraffin embedding.

### 2.10. Liver Histology

The paraffin sections (3–4 μm) were stained with hematoxylin–eosin (H&E) and examined under Olympus Provis AX70 microscope (Olympus Optical Co, Ltd, Tokyo, Japan). All the observations were carried out by an experienced pathologist.

### 2.11. Biochemical Assays

Approximately 0.1 g of liver tissue was homogenized with 2 mL of chloroform/methanol (2:1, *v*/*v*) solution and centrifuged at 3000 rpm for 10 min at room temperature. The lower clear layer was carefully transferred into a new test tube and dried by a stream of nitrogen. The dried lipids were re-dissolved in methanol and used for lipid analysis. The hepatic triglyceride (TG), total cholesterol (TC), and high-density lipoprotein (HDL) levels were analyzed with commercial kits according to the manufacturer’s protocol. Adiponectin and leptin were determined using ELISA kits (R&D Systems, USA). Aspartate aminotransferase (AST) and alanine aminotransferase (ALT) levels were measured using AST and ALT assay kits (Asanpharm, Seoul, Korea). The amount of nitric oxide (NO) in serum was determined by using the Griess reagent system (Promega, Madison, WI, USA) by following the manufacturer’s instructions. The determination of malondialdehyde (MDA) by thiobarbituric acid (TBA) was used as an index of lipid peroxidation [33]. Approximately 0.1 g of liver tissue was homogenized in 1 mL of 1.15% potassium chloride (Sigma-Aldrich, St. Louis, MO, USA). Then, 0.2 mL of 8.1% sodium dodecyl sulfate solution (SDS) (IBS-BS003a, iNtRON Biotechnology, Seongnam-si, Korea), 1.5 mL of 20 % acetic acid (Daejung Chemical, Suwon, Korea), 1.5 mL of 0.8% aqueous TBA (Sigma-Aldrich, St. Louis, MO, USA), and 0.7 mL of distilled water were added to the liver homogenate. The mixture was heated in a water-bath at 95 °C for 30 min. Thereafter, the heated mixture was immediately cooled in ice. One mL of distilled water and 5 mL of n-butanol (SAMCHUN, Seoul, Korea) were added in a consecutive order and centrifuged at 4000 rpm for 20 min (Combi-514R, Hanil Co. Ltd., Seoul, Korea). The absorbance of the clear supernatant was measured at 532 nm with an ELISA reader (Epoch Microplate Spectrophotometer, Biotek Inc., Winooski, VT, USA). Hepatic MDA content was expressed as nmol MDA per g of liver tissue. Standard curve was calculated using 1,1,3,3-tetraethoxypropane (Sigma-Aldrich, St. Louis, MO, USA). To quantify antioxidant activities, liver tissue was homogenized with PBS and centrifuged at 1500 g for 15 min. Subsequently, the clear supernatants were collected and used as samples. Hepatic SOD, GPx, and CAT were measured using a colorimetric assay kit (BlueGene Biotech, Shanghai, China) and the amount of nitric oxide (NO) was determined by using the Griess reagent system (Promega, Madison, WI, USA).

### 2.12. Western Blot Analysis of Liver Tissue

To examine the molecular mechanisms of HBE on triglyceride accumulation in liver, lipogenesis markers and AMPK and ACC phosphorylation were investigated by Western blot analysis. The liver tissue sample was extracted by protein lysis buffer (iNtRON Biotechnology, Seongnam-si, Korea) and subjected to western blot analysis using the same procedure as described in the HepG2 cells.

### 2.13. Data Analyses

All data are expressed as mean ± SEM. At least three separate experiments were performed, and each experiment was performed in triplicate. Statistical analysis was performed using GraphPad prism 5.03 (GraphPad Software, San Diego, CA, USA) with one-way ANOVA and Tukey’s post-hoc tests. Animal experiments analyses were carried out using Duncan’s multiple range test. p values less than 0.05 were considered to be statistically significant. All statistical analyses were conducted using software SPSS version 23 (SPSS Inc., Chicago, IL, USA). Experimental data are expressed as mean ± S.D. (*n* = 7 per group). Differences between mean values for individual groups were assessed by one-way analysis of variance (ANOVA) with Duncan’s multiple range test. *p* values less than 0.05 were considered to be statistically significant. *p* values of less than 0.05, 0.01, and 0.001 are described as * *p* < 0.05, ** *p* < 0.01, and *** *p* < 0.001, respectively.

## 3. Results

### 3.1. Cell Viability of HBE

The CCK-8 assay was used to demonstrate the effects of various concentrations of HBE plus 1 mM FFA on cell viability. HBE treatment at a concentration of up to 1000 μg/mL with 1 mM FFA for 24 h did not demonstrate a significant decrease in cell viability (Figure 2).

### 3.2. Effects of HBE on FFA-Mediated Steatosis

A significant increase in the lipid droplets was observed in 1 mM FFA-treatment group without added HBE compared with the control group (Figure 3A). However, at high concentrations (i.e., at 500 and 1000 μg/mL of HBE) along with 1 mM FFA treatment, the lipid droplets and lipid accumulation significantly decreased compared to the 1 mM FFA only group (Figure 3A,B). TG accumulation was induced by an exposure to 1 mM FFA treatment for 24 h. Intracellular TG content showed a significant decrease in the HBE treated HepG2 cells (Figure 3C).

### 3.3. Effects of HBE on Lipogenesis in FFA-Induced Steatosis

As shown in Figure 4A–D, the relative mRNA expression of SREBP-1c, C/EBPα, PPARγ, and FAS transcriptional factors was increased significantly in cells treated with 1mM FFA compared to untreated control cells. However, these mRNA expressions were significantly reduced in HBE- treated HepG2 cells. In addition, the protein level of lipogenic factors, SREBP-1c, C/EBPα, PPARγ, and FAS were also diminished by HBE in a concentration-dependent manner (Figure 4E). These results suggest that HBE attenuated FFA-induced steatosis through the down-regulation of lipogenesis.

### 3.4. Effects of HBE on Phosphorylation of AMPK and ACC and Immunoblotting of CPT-1 and PPARα in HepG2 Cells

FFA induced steatosis and decreased phosphorylation of AMP-activated Protein Kinase (AMPK) and Acetyl-CoA Carboxylase (ACC), which is one of the AMPK downstream targets. Following HBE treatment for 24 h, AMPK and ACC phosphorylation levels were significantly augmented compared to FFA treatment (Figure 5).

The mRNA expressions of CPT-1 and PPARα (related to β-oxidation) were consistently increased in a dose-dependent manner after HBE supplementation in FFA-induced HepG2 cells (Figure 5A,B).

### 3.5. Effects of HBE on Blood and Liver in HFD-fed Mice

As shown in Table 2, animals in all the groups gained body mass during the experimental period (Appendix A). The HFD-fed mice group gained more weight than the ND fed mice. The supplemented HBE (LH, 0.5% or MH, 1%) group witnessed markedly reduced body mass compared to the HFD control (Appendix A). However, there was no significant difference between LH (47.4 ± 1.4) and MH (48.2 ± 2.7). Interestingly, the ratio of organ weight to body mass was reduced after supplementation with HBE (LH, 0.5% or MH, 1%) treatments (Table 2, Figure 6A). In serum biochemical assays as shown in Table 2, HFD group exhibited significant increase in serum leptin, TG, TC, ALT, AST, and NO levels compared to those of HBE (LH, 0.5% or MH, 1%) treated groups (Table 2, Figure 6C). Serum MDA (an oxidative marker) levels were increased markedly by 89% in the HFD group compared to those of the control group. However, HBE-treated (LH, 0.5% or MH, 1%) groups showed a significant reduction in serum MDA levels than the HFD group (Figure 6D). As expected, HFD supplemented with 0.5% HBE (LH) and 1% HBE (MH) also revealed a dramatic improvement in serum and hepatic parameters in HFD-fed mice (Table 2).

### 3.6. Effects of HBE on the Hepatic Steatosis in HFD-Fed Mice

To determine the effect of HBE on protection of fat accumulation in liver of HFD mice, we performed a histological analysis. H&E staining of liver sections showed damage to hepatocytes with significantly large-sized lipid droplets in liver tissue of the HFD group (Figure 6A,B). However, the size of lipid droplets decreased in the liver tissue of the groups supplemented with HBE (LH, 0.5% or MH, 1%) (Figure 6B). Taken together, these results show that supplementation of HBE can alleviate hepatic steatosis in HFD-fed mice.

### 3.7. Effects of HBE on Hepatic Lipogenic Gene Expression of HFD-Fed Mice

To examine the molecular mechanisms of HBE on triglyceride accumulation in the liver, lipogenesis markers were investigated by real-time PCR and Western blot analysis. As shown in Figure 7A–D, the expression of SREBP-1c, C/EBPα, PPARγ, and FAS mRNA increased significantly in the HFD group compared with the NC group. In the HBE-supplemented (LH, 0.5% or MH, 1%) group, mRNA expression levels decreased significantly compared with those of HFD control. Moreover, Western blot analysis showed that HBE could inhibit the up-regulation of lipogenic factors. The results of Western blot analysis (Figure 7E) and real-time PCR suggests that HBE could have significantly diminished the expression of lipogenic genes, SREBP-1, C/EBPα, PPARγ, and FAS, of the HFD group in a dose-dependent manner.

### 3.8. Effects of HBE on Phosphorylation of AMPK and ACC and Immunoblotting of CPT-1 and PPARα in HFD-Fed Mice

Compared to the HFD group, AMPK and ACC phosphorylation were increased in a dose-dependent manner in the HBE-supplemented (LH, 0.5% or MH, 1%) group (Figure 8C,D). In addition, HBE treatment up-regulated CPT-1 and PPARα gene and protein expressions in the supplemented HBE (LH, 0.5% or MH, 1%) group compared with the HFD group (Figure 8A,B,E). In immunoblotting analysis of CPT-1 and PPARα, dose-dependent elevations were observed in response to HBE treatments.

## 4. Discussion

In this study, we investigated whether HBE supplementation ameliorates lipogenesis associated with NAFLD partly through the regulation of the expression of genes involved in lipid metabolism, using the HepG2 cell line and HFD-fed obese mice. To the best of our knowledge, this is a first study demonstrating that HBE can attenuate NAFLD *in vivo* and *in vitro*. TG accumulation was substantially reduced by ~77.42% in the 1% HBE supplemented group in HFD-fed obese mice. Blackberry, wild blueberry, strawberry, and chokeberry were also observed to have an inhibitory effect on oleic acid-induced TG accumulation in HepG2 cells [15]. In addition, both HBE and C3G showed increased insulin-secreting activity that led to diminished glucose production in INS-1 cells [18] and a dose-dependent hepatoprotective effect on tert-Butyl hydroperoxide (tBHP) induced HepG2 cell damage [19]. Takikawa et al. [34] showed that the anthocyanin-rich bilberry extract efficiently reduced hyperglycemia in type 2 diabetic mice via activation of AMPK in the liver, which consequently resulted in significantly decreased liver and serum lipid contents. These reports demonstrate that the anthocyanins or anthocyanin-rich extracts are regarded as indispensable agents in preventing NAFLD, in accordance with our results with honeyberry.

Recent large-scale studies indicated that purified anthocyanins or anthocyanin-rich extracts were associated with reduced risk of type 2 diabetes and cardiovascular diseases [35,36]. Results from *in vivo* and *in vitro* experiments demonstrated that anthocyanins may decrease hyperglycemia and improve lipid profile, thus protecting against NAFLD-related metabolic alterations mediated via activation of AMPK [17,22,37]. In addition, antioxidant and anti-inflammatory effects of anthocyanins have been explored for therapeutic efficacy in numerous diseases [38,39].

In our study, the expressions of SREBP-1c, C/EBPα, PPARγ, and FAS changed after treatment of FFA-treated HepG2 cells or HFD-fed mice with HBE (Figure 4 and Figure 8). Supplementation of HBE led to higher phosphorylation of AMPK and ACC; thus, the expression of lipogenesis markers diminished and TG accumulation was further suppressed. These results demonstrate that HBE might have beneficial effects on hepatic steatosis by targeting the AMPK signaling pathway. AMPK directly impacts lipid metabolism by inhibiting fat accumulation through the modulation of several downstream-signaling components such as ACC [40]. Phosphorylation of ACC leads to interruption of fatty acid synthesis, and ACC inhibition could also increase β-oxidation by promoting the activity of CPT-1, which is essential for the entry of long-chain fatty acids into mitochondria [41]. In this study, HBE significantly up-regulated the expression of CPT-1, affecting fatty acid oxidation in HepG2 cells and HFD-fed mice (Figure 5B and Figure 8A). Accumulation of fatty acids is toxic for several tissues, especially the liver, because it alters metabolism [42]. Our results indicated that HBE-induced activation of AMPK and ACC are critical regulators of the AMPK signaling pathway, which act by controlling hepatocellular lipid metabolism. Additionally, HBE (LH, 0.5% or MH, 1%) supplementation induced activation of AMPK and expression of CPT-1 and PPARα, and consequently suppressed lipid accumulation by activating AMPK [43] in HFD-fed mice.

AMPK is a master regulator of cellular metabolism and responsible for overall energy balance [44]. AMPK regulates cellular lipid metabolism through direct phosphorylation of ACC and fatty acid oxidation by the CPT-1. In addition, AMPK inhibits transcription factor, SREBP1, a chief transcriptional regulator of lipid synthesis and FAS proteins [24,44].

On the whole, we provide evidence to prove that supplemented HBE suppressed TG and cholesterol synthesis and reduced cellular lipid accumulation by increasing AMPK and ACC phosphorylation, thereby inhibiting expression of SREBP-1c, FAS, and PPARγ. Taken together, our results suggest that HBE might have beneficial effects on hepatic steatosis by targeting the hepatic AMPK-mediated fatty acid metabolism. Our findings highlight that HBE may improve hepatic steatosis and ameliorate HFD-induced fatty liver disease. Therefore, we propose HBE as a potential therapeutic agent for NAFLD.

## 5. Conclusions

NAFLD is the most common liver disorder. Excessive accumulation of TG due to esterification of glycerol and FFA induces the NAFLD. This study showed that HBE effectively reduced TG accumulation through down-regulation of hepatic lipid metabolic gene expression and up-regulation of the activation of AMPK and ACC signaling in both the HepG2 cells as well as in livers of diet-induced obese mice. HBE might have beneficial effects on hepatic steatosis by targeting the hepatic AMPK-mediated fatty acid metabolism. 

## Figures and Tables

**Figure 1 nutrients-11-00494-f001:**
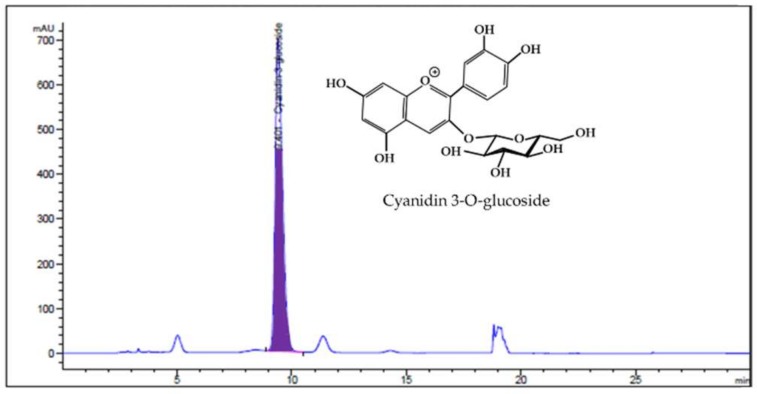
High-Performance Liquid Chromatography with Diode Array Detection (HPLC-DAD) chromatogram of main component in honeyberry (*Lonicera caerulea*). HPLC-DAD traces at 517 nm, Cyanidin 3-O-glucoside (C3G), was detected at 5.91 mg/mL in 25% ethanolic extract of honeyberry using high-performance liquid chromatography (HPLC) system.

**Figure 2 nutrients-11-00494-f002:**
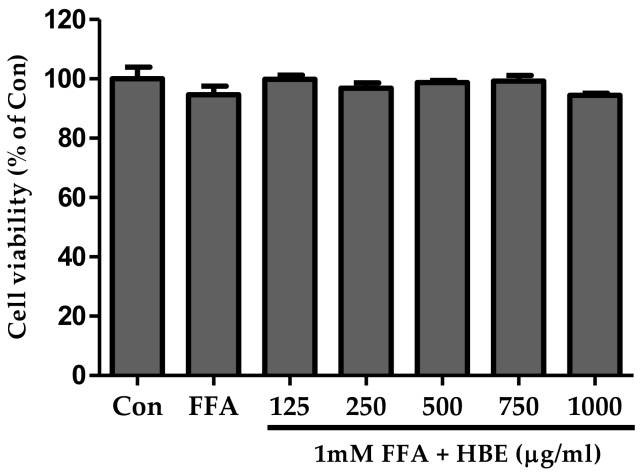
Effect of Honeyberry extract (HBE) on the viability of HepG2 cells. HepG2 cells were incubated in the various concentration of HBE with 1 mM free fatty acids (FFA) for 24 h. All experiments were repeated at least three times and data represent means ± SD.

**Figure 3 nutrients-11-00494-f003:**
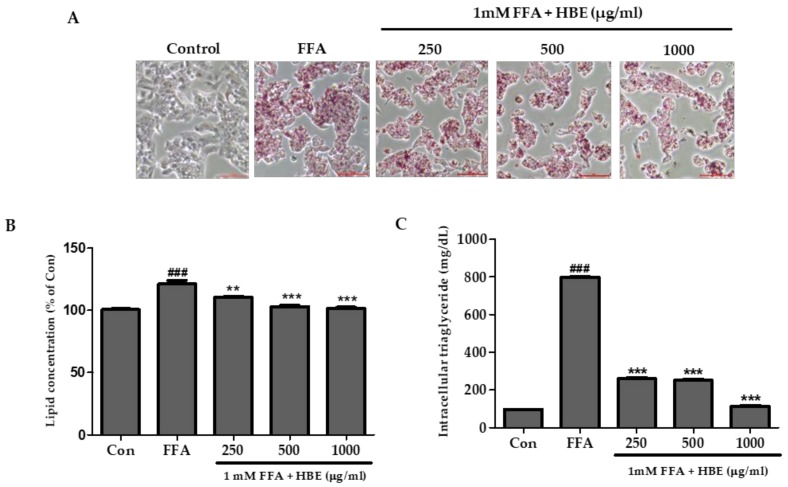
Effect of honeyberry extract (HBE) on Oil Red O staining and lipid accumulation in HepG2 cells. Lipid droplets in HepG2 cells were dyed red (magnification 200X). (**A**) Oil Red O staining images of HepG2 cells treated with 1 mM free fatty acids (FFA) and exposed to various concentration of HBE with 1mM FFA for 24 h. Control (Con) cells were incubated with 1% fat-free bovine serum albumin. (**B**) Quantitative lipid accumulation of Oil Red O contents at 500 nm. (**C**) Total intracellular triglyceride in HepG2 cells treated with HBE and 1 mM FFA. Data represent means ± SD. ### *p* < 0.001 vs. Con; ** *p* < 0.01, *** *p* < 0.001 vs. FFA.

**Figure 4 nutrients-11-00494-f004:**
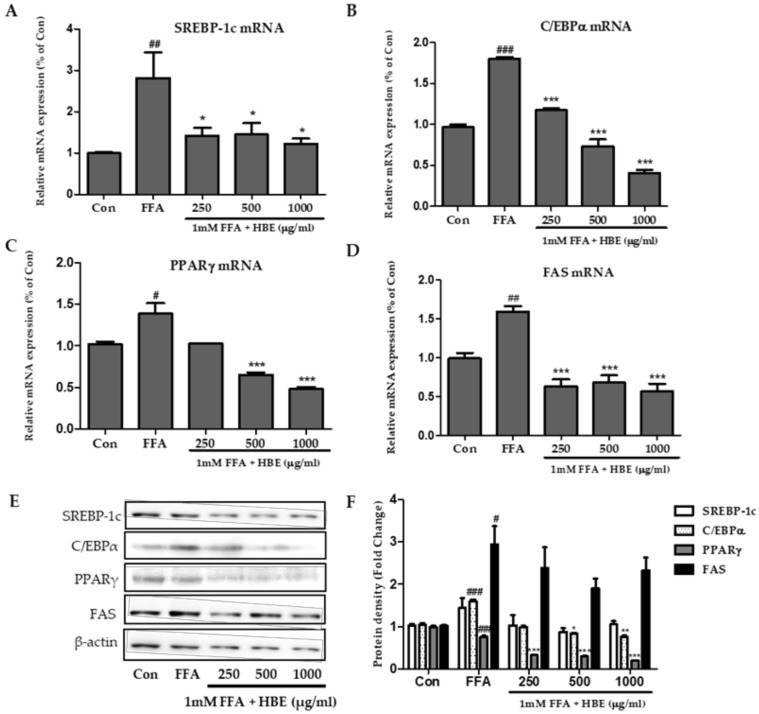
Effects of HBE on the expression of genes associated with lipogenesis in HepG2 cells. (**A**–**D**) The expressions of SREBP-1c (**A**), C/EBPα (**B**)*,* PPARγ (**C**), and fatty acid synthase (FAS) (**D**) were quantified by real-time PCR and normalized by β-actin as an internal control. (**E**) SREBP-1c, C/EBPα, PPARγ, and FAS protein levels were monitored by Western blot analysis. (**F**) Protein density of SREBP-1c, C/EBPα, PPARγ, and FAS. Equal loading of protein was verified by probing β-actin. Data represent means ± SD. # *p* < 0.5, ## *p* < 0.01, ### *p* < 0.001 vs. Con; * *p* < 0.05, ** *p* < 0.01, *** *p* < 0.001 vs. free fatty acids (FFA).

**Figure 5 nutrients-11-00494-f005:**
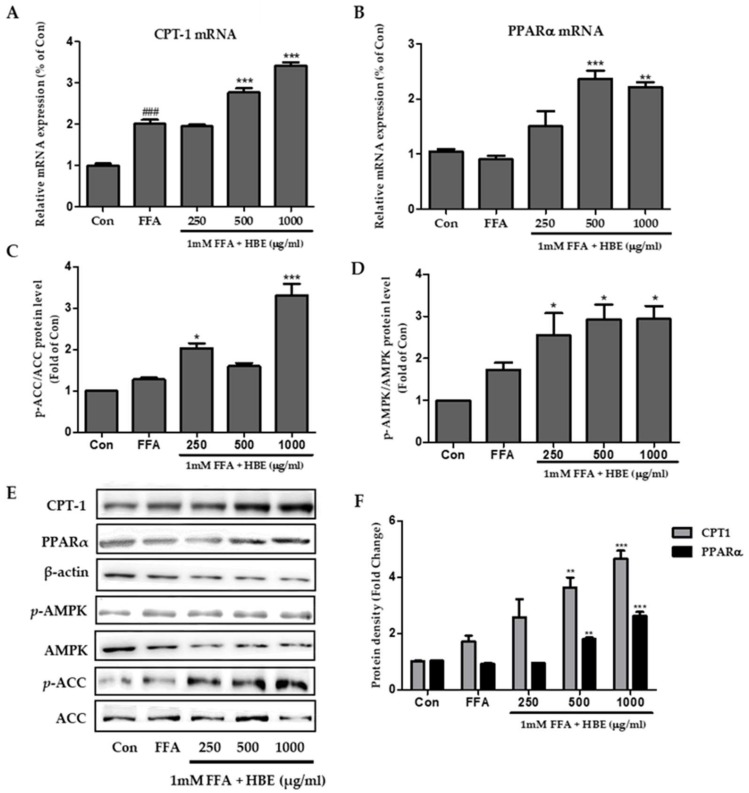
Effects of HBE on CPT1 and PPARα expression and AMPK, ACC signaling in FFA-treated HepG2 cells, with and without HBE supplementation. The mRNA expression of genes associated with fatty acid oxidation factors CPT1 (**A**) and PPARα (**B**) were quantified by real-time PCR and normalized by β-actin as an internal control. Western blot analysis of p-AMPK/AMPK (**C**) and p-ACC/ACC (**D**) in HepG2 cells. AMPK and ACC were used as a protein loading control of phosphorylated AMPK (p-AMPK) and phosphorylated ACC (p-ACC), respectively. (**E**) CPT-1, PPARα, phosphorylated AMPK and phosphorylated ACC proteins level by immunoblot analysis. The results from three independent experiments are expressed as the mean ± SD. (**F**) Protein density of CPT-1 and PPARα. Equal loading of protein was verified by probing β-actin. The results from three independent experiments are expressed as the mean ± SD. ### *p* < 0.001 vs. Con; ∗ *p* < 0.05, ** *p* < 0.01, *** *p* < 0.001 vs. FFA.

**Figure 6 nutrients-11-00494-f006:**
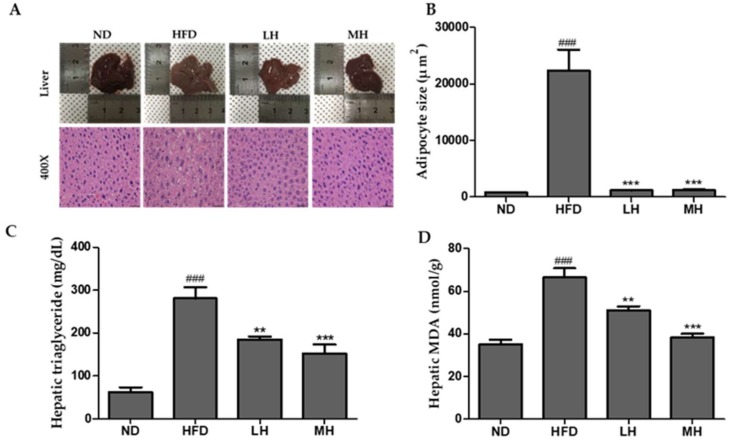
Effect of HBE treatment on hepatic steatosis in normal or HFD-fed mice. (**A**) Liver tissue histology (400×). (**B**) Adipocyte sizes of ND, HFD, LH, and MH. (**C**) The accumulation of liver TG. (**D**) Hepatic MDA levels. ND, normal diet control; HFD, high-fat diet; LH, HFD-supplemented 0.5% HBE; MH, HFD-supplemented 1% HBE. The results from three independent experiments are expressed as the mean ± SD. ### *p* < 0.001 vs. ND; ** *p* < 0.01, *** *p* < 0.001 vs. HFD.

**Figure 7 nutrients-11-00494-f007:**
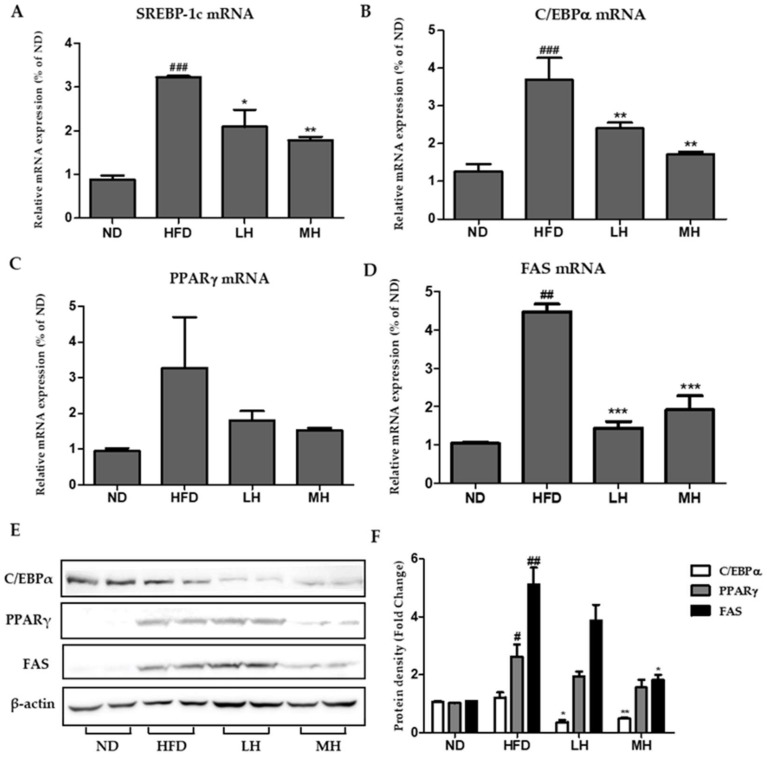
Effects of HBE on the expression of genes associated with lipogenesis in HFD-fed mice. (**A**–**D**) The expression of SREBP-1c (**A**), C/EBPα (**B**)*,* PPARγ (**C**), and FAS (**D**) were quantified by real-time PCR and normalized by β-actin as an internal control. (**E**) C/EBPα, PPARγ, and FAS protein levels by Immunoblot analysis. (**F**) Protein density of C/EBPα, PPARγ, and FAS. Equal loading of protein was verified by probing β-actin. ND, normal diet control; HFD, high-fat diet; LH, HFD-supplemented 0.5% HBE; MH, HFD-supplemented 1% HBE. Data represent means ± SD. # *p* < 0.5, ## *p* < 0.01, ### *p* < 0.001 vs. ND; * *p* < 0.5, ** *p* < 0.01, *** *p* < 0.001 vs. HFD.

**Figure 8 nutrients-11-00494-f008:**
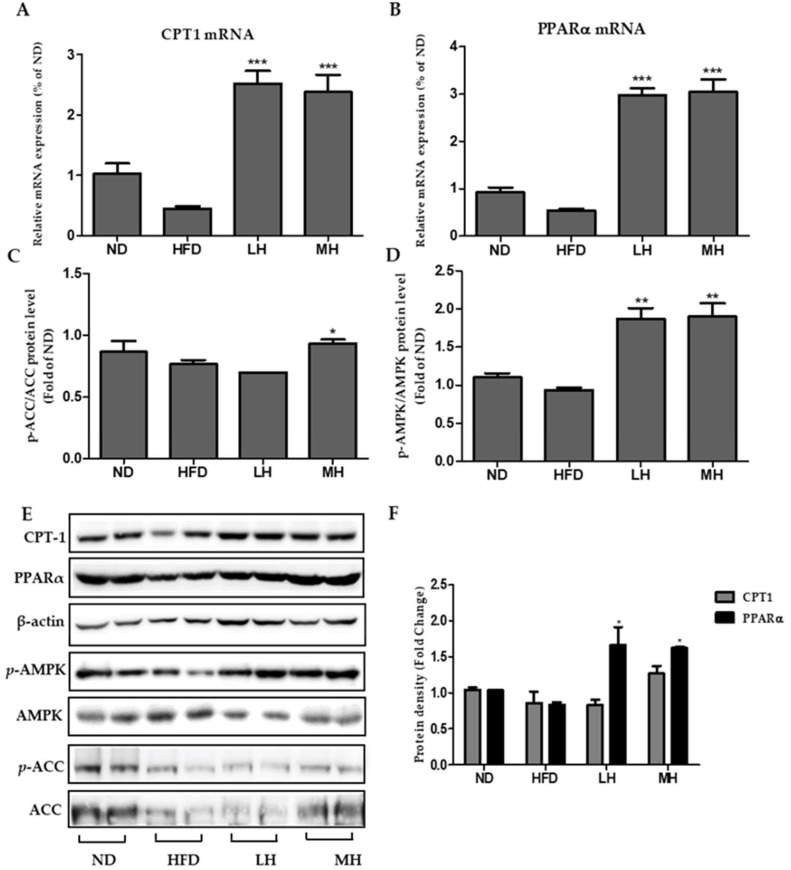
Effects of HBE on CPT-1 and PPARα expression and AMPK, ACC signaling in HFD-fed mice. The expression of CPT-1 (**A**) and PPARα (**B**) were quantified by real-time PCR and normalized by β-actin as an internal control. (**C**–**D**) Western blot analysis of p-AMPK/AMPK and p-ACC/ACC protein in the livers of mice fed an ND, HFD, or HFD with supplemented HBE (LH, 0.5% or MH, 1%). β-actin was used as a protein loading control. AMPK and ACC were used as protein loading controls of phosphorylated AMPK (p-AMPK) and phosphorylated ACC (p-ACC), respectively. (**E**) CPT-1, PPARα, phosphorylated AMPK, and phosphorylated ACC protein levels by immunoblot analysis. (**F**) Protein density of CPT-1, PPARα. Equal loading of protein was verified by probing β-actin. Data represent means ± SD. * *p* < 0.5, ** *p* < 0.01, *** *p* < 0.001 vs. HFD.

**Table 1 nutrients-11-00494-t001:** Primer sequence used for real-time quantitative PCR.

	Gene	Forward (5′–3′)	Reverse (5′–3′)
HepG2	PPARγ	TGCAGGTGATCAAGAAGACG	AGTGCAACTGGAAGAAGGGA
	C/EBPα	TGGACAAGAACAGCAACGAGTA	ATTGTCACTGGTCAGCTCCAG
	SREBP-1c	GCGCCTTGACAGGTGAAGTC	GCCAGGGAAGTCACTGTCTTG
	FAS	CCCCTGATGAAGAAGGATCA	ACTCCACAGGTGGGAACAAG
	CPT-1	CCTCCGTAGCTGACTCGGTA	GGAGTGACCGTGAACTGAAA
	PPARα	ACGATTCGACTCAAGCTGGT	GTTGTGTGACATCCCGACAG
	β-actin	CTCTTCCAGCCTTCCTTCCT	AGCACTGTGTTGGCGTACAG
Mouse Liver	PPARγ	CAGGAGAGCAGGGATTTGCA	CCTACGCTCAGCCCTCTTCAT
	C/EBPα	TTACAACAGGCCAGGTTTCC	GGCTGGCGACATACAGTACA
	SREBP-1c	ATCGCAAACAAGCTGACCTG	AGATCCAGGTTTGAGGTGGG
	FAS	TTGCTGGCACTACAGAATGC	AACAGCCTCAGAGCGACAAT
	CPT-1	CTCAGTGGGAGCGACTCTTCA	GGCCTCTGTGGTACACGACAA
	PPARα	CAGGAGAGCAGGGATTTGCA	CCTACGCTCAGCCCTCTTCAT
	β-actin	CTGTCCCTGTATGCCTCTG	ATGTCACGCACGATTTCC

PPARγ, peroxisome proliferator-activated receptor gamma; C/EBPα, CCAAT/enhancer-binding protein alpha; SREBP-1c, sterol regulatory element-binding protein1-c; FAS, fatty acid synthase; CPT-1, carnitine palmitoyltransferase-1; PPARα, Peroxisome proliferator-activated receptor alpha.

**Table 2 nutrients-11-00494-t002:** Effects on blood and hepatic parameters in HFD-fed mice receiving 6 weeks of treatments.

Groups	ND	HFD	LH	MH
Food intake (g/day)	3.8 ± 0.1 ^ns^	4.0 ± 0.1	3.9 ± 0.5	3.9 ± 0.3
FER	0.020 ± 0.00 ^b^	0.025 ± 0.00 ^a^	0.022 ± 0.00 ^ab^	0.022 ± 0.00 ^ab^
Body mass (g)				
Initial (A)	23.1 ± 0.6 ^ns^	22.8 ± 1.6	22.3 ± 1.6	22.5 ± 1.0
Final (B)	45.9 ± 3.7 ^c^	53.7 ± 3.0 ^a^	47.4 ± 1.4 ^bc^	48.2 ± 2.7 ^bc^
Gains (B-A)	22.7 ± 3.6 ^c^	30.9 ± 2.6 ^a^	25.1 ± 0.2 ^b^	25.7 ± 1.7 ^b^
Relative organ weight (%)				
Liver	3.6 ± 0.3 ^ns^	4.0 ± 1.6	3.8 ± 0.2	3.8 ± 0.4
Abdominal adipose tissue	0.9 ± 0.3 ^b^	1.7 ± 0.4 ^a^	1.3 ± 0.4 ^b^	1.2 ± 0.7 ^b^
Epdidymal adipose tissue	4.1 ± 0.3 ^b^	5.3 ± 0.7 ^a^	4.6 ± 0.5 ^b^	4.4 ± 0.4 ^b^
Serum biochemical assays				
Adiponectin (ng/mL)	4.7 ± 0.6 ^a^	3.1 ± 1.3 ^b^	4.1 ± 1.2 ^ab^	4.3 ± 1.2 ^ab^
Leptin (pg/mL)	0.4 ± 0.2 ^c^	1.5 ± 0.4 ^a^	0.9 ± 0.2 ^b^	1.0 ± 0.4 ^b^
TG (mg/dL)	148.6 ± 11.0 ^b^	184.2 ± 30.4 ^a^	171.4 ± 36.7 ^ab^	142.7 ± 20.9 ^b^
TC (mg/dL)	163.6 ± 23.2 ^ab^	191.3 ± 42.2 ^a^	158.4 ± 15.7 ^b^	179.9 ± 18.2 ^ab^
HDL (mg/dL)	83.5 ± 6.1 ^a^	71.2 ± 7.4 ^b^	82.1 ± 7.2 ^a^	84.2 ± 9.3 ^a^
ALT (Karmen/mL)	18.3 ± 2.1 ^b^	29.1 ± 7.5 ^a^	22.8 ± 8.6 ^ab^	19.9 ± 5.6 ^b^
AST (Karmen/mL)	54.0 ± 3.3 ^b^	65.1 ± 7.7 ^a^	55.4 ± 6.1 ^b^	56.8 ± 10.4 ^b^
NO (µM)	9.5 ± 2.3 ^b^	26.1 ± 9.4 ^a^	17.9 ± 6.3 ^b^	15.8 ± 5.0 ^b^
Hepatic parameters				
MDA (nmol/g)	35.1 ± 5.8 ^c^	66.5 ± 11.4 ^a^	51.1 ± 4.7 ^b^	38.4 ± 4.4^c^
NO (µM)	38.9 ± 5.5 ^b^	53.1 ± 10.7 ^a^	42.0 ± 7.4 ^b^	41.9 ± 6.7 ^b^
SOD (ug/mL)	47.1 ± 1.9 ^b^	41.3 ± 2.2 ^b^	54.8 ± 11.5 ^a^	55.5 ± 4.1 ^a^
GPx (ug/mL)	1.4 ± 0.2 ^a^	1.0 ± 0.2 ^b^	1.3 ± 0.2 ^a^	1.3 ± 0.3 ^a^
CAT (ng/mL)	25.2 ± 3.1 ^a^	21.1 ± 1.0 ^b^	25.3 ± 5.7 ^a^	26.2 ± 2.7 ^a^

Data represent means ± SEM (n = 7 for each group). FER, food efficiency ratio; ns, not significant; TG, triglyceride; TC, total cholesterol; HDL, high density lipoprotein; ALT, alanine aminotransferase; AST, aspartate transaminase; NO, nitric oxide; MDA, malondialdehyde; SOD, superoxide dismutase; GPx, glutathione peroxidase; CAT, catalase; ND, normal diet control; HFD, high-fat diet, LH, HFD+0.5% HBE; MH, HFD+1% HBE. The different letters (a > b > c) within a column indicate significant differences (*p* < 0.05) determined by Duncan’s multiple range test.

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
