# Peer review of "Lonicera caerulea Extract Attenuates Non-Alcoholic Fatty Liver Disease in Free Fatty Acid-Induced HepG2 Hepatocytes and in High Fat Diet-Fed Mice"

_nutrients, 2019, doi:10.3390/nu11030494_

Reviewer 1 Report

In the present study, Park M. et al. have “investigate whether Lonicera caerulea extract (HBE) improved nonalcoholic fatty liver disease (NAFLD) on free fatty acid (FFA) induced HepG2 hepatocytes and mice fed high-fat diet (HFD) and the possible mechanisms involved”. The authors describe that HBE attenuates NAFLD in vivo and vitro: i) HBE supressed fatty acid synthesis and reduced lipid accumulation in HepG2 cells, and also ii) HBE decreased lipid accumulation in the liver of HFD-fed obese mice. Down-regulation of the hepatic lipid metabolic gene expression and up-regulation of the activation of the AMPK and ACC signalling might explain the benefical mechanism shown by HBE treatment.

Although this is a quite interesting study, it lacks further investigation to support the potencial protective role of HBE on NAFLD. Please find below major and minor comments associated with this manuscript.

Major comments:

1.    The aim should be rewrite.

2.    I do consider essential to include in the materials and methods section:

a.         Preparation of FFA

b.        HBE concentration used in cell culture experiments.

c.         Any reference of kit used for cell citotoxicity studies

d.        Western blot analysis should be rewrite (details of dilution of antibodies should be included)

e.         Composition of diets

f.          How have determined the hepatic parameters (MDA, SOD, GPx and CAT)?.

3.    I do consider essential to include the treatment with HBE in mice fed with normal diet.

4.    There are some issues in the result section that could be improved. I have some specific comments:

a.         Where is the Figure 1?

b.        Cell culture experiments included only as control group cells treated with 1% BSA; however, the authors used isopropyl alcohol for to prepare FFA. It is should be included this group as control group.

c.         Why the authors showed the lipid and TG concentration as % of control?.

d.        Why the WB experiments show only the blot?. It is necessary to include the cuantitative analysis.

e.         Statistical analyses of table 2 and figure 8 should be explain.

f.          b-actin blot in figure 5?

g.        The weight of adipose tissues and liver should be normalized by tibia length

5.    In the result section the authors state that “These data suggests that HBE supplemented induced activation of AMPK and expressions of CPT-1 and PPARa, suppress lipid accumulation through activation of AMPK on HFD fed mice”. To stablish and/or confirm this associations it would be necessary to perform a correlation between AMPK, CPT-1 and PPARa expression and lipid values.

Minor comments:

1.      The authors should include an explanation of superscripts a/b/c in legends of table 2.

2.      The authors should include the scheme of animal experimental protocol

3.      The authors should include the name of antibiotics used in cell culture

Author Response

 Dear reviewer,

Thank you for your letter and constructive comments concerning our manuscript. We have studied your comments carefully and made major correction. We answered your questions or comments in details in the following texts.

Major comments:

The aim should be rewrite.

à Thank you for your thoughtful comment. We missed writing important information. We rewrote the aim of this study in lines 76-79.

2.    I do consider essential to include in the materials and methods section:

a.         Preparation of FFA

à We prepared a conjugated FFA (OA:PA at 2:1) with sodium PA (Sigma, St. louis, MO, USA) and OA (Alfa Aesar, Haverhill, Massachusetts, USA) Palmitic acid was dissolved in isopropanol (stock solution 500 mM). We used final concentrations: PA 0.33 mM + OA 0.66 mM (final fatty acids concentration 1 mM) prepared in culture medium containing 1% bovine serum albumin (BSA). We have added most widely used reference for this step.

Ricchi et al., (J Gastroenterol Hepatol. 2009 May;24(5):830-40. doi: 10.1111/j.1440-1746.2008.05733.)

b.        HBE concentration used in cell culture experiments.

à We have added the HBE concentration on the line 107-108 as the followings.

 We prepared a HBE stock solution (20 mg/ml) and used 250, 500, and 1000 μg/ml in DMEM medium.

c.         Any reference of kit used for cell citotoxicity studies

à Thank you for your kind advice. We have added references for cell cytotoxicity measurement on the line 111-118 as the following:

Effects of HBE and FFA on cell proliferation were determined using Cell Counting Kit-8 (CCK-8) (Dojindo Molecular Technologies, USA) [31]. HepG2 cells were seeded in a 96-well plate at a concentration of 5 X 104 cells per well and then allowed to attach after 24 h. The medium was replaced with serum-free DMEM containing 1mM FFA and 1% BSA, with and without HBE for 24 h. The cells were incubated with 10 μl of CCK-8 and serum-free DMEM into each well and incubated at 37 for 2h. The experiment was performed in triplicate. The results were expressed as percentage of viable cells with respect to the untreated control cells. The absorbance was measured at 450 nm using a microplate spectrophotometer system (BioRad, USA) [32].”

< principle of the product CCK-8>

Reference [31]: Tominaga, H.; Ishiyama, M.; Ohseto, F.; Sasamoto, K.; Hamamoto, T.; Suzuki, K.; Watanabe, M. A water-soluble tetrazoliumsalt useful for colorimetric cell viability assay. Analytical Communications 1999, 36, 47-50, doi:10.1039/A809656B.

<< span=""> cell cytotoxicity measurement>

Reference [32]: Takanezawa, Y.; Nakamura, R.; Sone, Y.; Uraguchi, S.; Kiyono, M. Atg5-dependent autophagy plays a protective role against methylmercury-induced cytotoxicity. Toxicology Letters 2016, 262, 135-141, doi:https://doi.org/10.1016/j.toxlet.2016.09.007.

d.        Western blot analysis should be rewrite (details of dilution of antibodies should be included)

à  According to your comments, we have added the detailed information about antibodies on Materials and Methods section on the line 146-151 as the followings:

PVDF Membranes were immunoblotted with the primary antibodies C/EBPα (1:500), PPARγ (1:1000), SREBP-1c (1:1000), FAS (1:100), PPARα (1:1000), ACC (1: 2000), pACC (1: 10,000), pAMPK-α1 (1:1000), AMPK-α1 (1:1000), and CPT1 (1:1000) at 4 °C overnight. After three washes, the membranes were incubated with the anti-mouse or anti-rabbit horseradish peroxidase (HRP)-coupled secondary antibodies (1:5000) for 1 h at room temperature in 5% skim milk. All the band-density quantifications were normalized with β-actin.

e.         Composition of diets

à According to your comments, we have added a supplementary figure 1, describing diet composition.

AIN-93G, D10012G, Research diet: 7% fat

D12451, Research diet: HFD, HFD+0.5%, HFD+1%: Diet composition was described in supplement figure 1.

f.          How have determined the hepatic parameters (MDA, SOD, GPx and CAT)?.

à Thank you for your valuable comments. According to your advice, we have added hepatic parameters information on Materials and Methods section on the line 188-204 as the followings

 The determination of malondialdehyde (MDA) by thiobarbituric acid (TBA) was used as an index of lipid peroxidation [31]. Approximately 0.1 g of liver tissue was homogenized in 1 mL of 1.15 % potassium chloride (Sigma-Aldrich, USA). Then, 0.2 mL of 8.1% sodium dodecyl sulfate solution (SDS) (IBS-BS003a, iNtRON Biotechnology, Korea), 1.5 mL of 20 % acetic acid (Daejung Chemical, Korea), 1.5 mL of 0.8 % aqueous TBA (Sigma-Aldrich, USA), and 0.7 mL of distilled water were added to the liver homogenate. The mixture was heated in water bath at 95 °C for 30 min. Thereafter, the heated mixture was cooled in ice, immediately. 1 mL of distilled water and 5 mL of n-butanol (SAMCHUN, Korea) were added in consecutive order and centrifuged at 4,000 rpm for 20 min (Combi-514R, Hanil Co.Ltd., Korea). The absorbance of clear supernatant was measured at 532 nm with ELISA reader (Epoch Microplate Spectrophotometer, Biotek Inc., USA). Hepatic MDA content was expressed as nmol MDA per g of liver tissue. Standard curve was calculated with 1,1,3,3-tetraethoxypropane (Sigma-Aldrich, USA). To quantify antioxidant activities, liver tissue was homogenized with PBS and centrifuged at 1,500 g for 15 min. Then, clear supernatant was collected and used for samples. Hepatic SOD, GPx, and CAT were measured using a colorimetric assay kit (BlueGene Biotech, China) and the amount of nitric oxide (NO) was determined by using the Griess reagent system (Promega, USA).

3.    I do consider essential to include the treatment with HBE in mice fed with normal diet.

  à Thank you for your comments. Our aim of the study was to examine the effect of HBE against non-alcoholic liver disease. To briefly provide an explanation of our study model (Supplementary Fig 1), we gave high fat diet to induce obese mice and then treated HBE on those obese mice. According to reports, NAFLD is generally induced by obesity and thus high fat-induced obese mice is likely to develop NAFLD while normal diet fed mice is not likely to develop NAFLD1,2,3). Normal diet fed mice weight gain was significantly lower than high fat-induced mice.

1) Zhou et al., Biomed Pharmacother. 2018 Jan;97:1397-1408. doi: 10.1016/j.biopha.2017.10.035. 

2) Chen et al., Food Funct. 2019 Jan 24. doi: 10.1039/c8fo01236a.

3) Canfora et al., 2019 (Nat Rev Endocrinol. 2019 Jan 22. doi: 10.1038/s41574-019-0156-z.)

Cimini et al. (World J Gastroenterol. 2017 May 21; 23(19): 3407–3417.) reported that NAFLD rises dramatically in high-risk individuals such as patients with diabetes (60%), hyperlipidemia (90%) and obese patients (91%).

Because the purpose of our study was to examine the effect of HBE on NAFLD, we only used obese mice and did not give HBE to normal diet fed mice. Normal diet fed mice was used for comparison purposes against high fed-induced obese mice to check if high fat fed mice weight gain was significantly larger than weight gain of normal diet fed mice, because only then we would be able to evaluate the effect of HBE on NAFLD through obese mice.

4.    There are some issues in the result section that could be improved. I have some specific comments:

a.         Where is the Figure 1?

à It is located in Materials and Methods section on the line 98-100.  

2.2 Analysis of HBE by HPLC.

Figure 1 is HPLC-DAD chromatogram of main component in honeyberry (Lonicera caerulea).

b.        Cell culture experiments included only as control group cells treated with 1% BSA; however, the authors used isopropyl alcohol for to prepare FFA. It is should be included this group as control group.

à Thank you for your insightful comment. We used isopropyl alcohol to dissolve FFA and thus isopropyl alcohol was used as a solvent. Because cells with FFA-mediated steatosis already include isopropyl alcohol, we did not include a control group (only adding isopropyl alcohol to cells) when treating the FFA-mediated steatosis cells with HBE. There is similar in vitro model we used like below: We treated 1 mM FFA in DMEM media with 1% BSA (references Ricchi et al., Lee et al., and Takahara et al.)

Ricchi et al. (Journal of Gastroenterology and Hepatology 2009, 24, 830-840, doi:doi:10.1111/j.1440-1746.2008.05733.x.)

Lee et al. (Phytomedicine. 2018 Jul 18;55:14-22. doi:10.1016/j.phymed.2018.07.008.).

Takahara et al., (PLoS One. 2017 Mar 9;12(3):e0170591. doi: 10.1371/journal.pone.0170591. eCollection 2017.)

Three studies used 1mM FFA to induce steatosis in HepG2 cell and FFA stock solution was prepared in isopropanol or methanol. But they used 1% BSA only as control group. In our study, we prepared 500mM FFA stock solution for in vitro model and used 1mM FFA in DMEM media.

c.         Why the authors showed the lipid and TG concentration as % of control?.

à Thank you for your insightful comments. In figure 3, we have changed % to concentration in Figure 3C (B instead of A) as your comment. However, Result of Oil-Red O staining is always shown as % of control.

à Control was set as 100%                              à Result (mg/dL)

d.        Why the WB experiments show only the blot?. It is necessary to include the cuantitative analysis.

à Thank you for your insightful comments. We have added quantitative analysis of western blot in Figure 4 (F), Figure 5 (F), Figure 7 (F), and Figure 8 (F).

e.         Statistical analyses of table 2 and figure 8 should be explain.

à Statistical analyses of all data were described in Materials and Methods section on the line 211-220. In the footnote of table 2, different superscript (a, b, c) indicate that the results have significant differences (p<0.05). For example, value with superscript “a” and value with superscript “b” are significantly different. Values with same superscript are not statistically different. We believe that these letters well-represent statistically significant differences. The statistic method for Fig 8 is equal to Fig 7. the western blot data.

f.          b-actin blot in figure 5?

à Thank you for your kind comments. We have added β-actin blot in figure 5 and figure 8.

g.        The weight of adipose tissues and liver should be normalized by tibia length

à Thank you for your comments. We are very sorry that we did not measure tibia length at sacrifice. We realize that tibia length is very useful to estimate stature from dry bone length or from the size of other body parts. We wish to consider tibia length in further study.

5.    In the result section the authors state that “These data suggests that HBE supplemented induced activation of AMPK and expressions of CPT-1 and PPARa, suppress lipid accumulation through activation of AMPK on HFD fed mice”. To stablish and/or confirm this associations it would be necessary to perform a correlation between AMPK, CPT-1 and PPARa expression and lipid values.

 à Thank you for your comments. The phrase ‘suppress lipid accumulation of PPARα, through activation of AMPK’ is not our result. By mistake, it is part of discussion and came from the study results of Pawlak et al. (J Hepatol. 2015 Mar;62(3):720-33. doi: 10.1016/j.jhep. Review). We removed this phrase from the result. 

Minor comments:

The authors should include an explanation of superscripts a/b/c in legends of table 2.

à Thank you for your kind comments. All statistical analyses were conducted using software SPSS version 23 (SPSS Inc., Chicago, USA). Experimental data are expressed as mean ± S.D. (n = 7 per group). Differences between mean values for individual groups were assessed by one-way analysis of variance (ANOVA) with Duncan’s multiple range test. Statistically significant difference was considered at p < 0.05. ‘The different letters (a > b > c) within a column indicate significant differences (p < 0.05) determined by Duncan’s multiple range test’. This is included in legends of table 2. We believe that these letters well-represent statistically significant differences between individual groups. ex). Means with same superscript are not statistically different from each other.

The authors should include the scheme of animal experimental protocol

à Thank you for your comments. We have added a supplementary figure 2 about scheme of animal experimental protocol.

The authors should include the name of antibiotics used in cell culture

à Thank you for your kind comments. We have added antibiotics name in Materials and Methods.

Reviewer 2 Report

Dear Authors,

Thank you for the opportunity to review your manuscript. Please see below for specific comments on each section.

Abstract:

-Please remove the lines space before '...As a result...' Please also delete this phrase.

Introduction:

-Lines 51-52: Please revise the sentence, 'It is characterized by the ability to survive in the wild without care during growing growth [12]'.

-Line 54: Please specify the health benefits.

-Lines 75-76: Please provide more detail on the study aims, in terms of the specific variables under investigation and units of measure.

Methods:

-Lines 80-85: Please provide references to justify the methods used for the preparation of HBE.

-Lines 147-148: Please specify details on the quantification of food intake and body mass (e.g., the time of day for the measurement). Please replace 'body weight' with 'body mass'.

Results:

-Line 181: Please replace the word 'substantial' with 'significant' if there was in fact no significant difference.

-Line 179: Please re-phrase this sentence so that it begins with, 'The CCK-8 assay indicated that...'

-Lines 186-188, and lines 196-197: Please move this paragraph to the Methods.

-Lines 235-238: Please move the text in these lines to the Discussion.

-Lines 240-241, lines 261-262, lines 275-276, and lines 292-293: Please move the text in these lines to the Methods.

-Lines 298-300: Please move this text to the Discussion.

Discussion:

-Line 310: It is suggested that the Discussion begins with a statement of findings of the current study, rather than referring to findings of previous studies.

-Line 341: Please provide a reference for this statement.

Author Response

Dear reviewer,

Thank you for your letter and constructive comments concerning our manuscript. We have studied your comments carefully and made major correction. Based on your feedback, we answer your questions or comments in details in the following texts.

Abstract:

-Please remove the lines space before '...As a result...' Please also delete this phrase.

à According to your advice, we have deleted this phrase and removed the lines space before ‘HBE suppressed fatty acid…’.

Introduction:

-Lines 51-52: Please revise the sentence, 'It is characterized by the ability to survive in the wild without care during growing growth [12]'.

à Line 51-52: Thank you for your thoughtful advice, we have revised the sentence, ‘Its fruits are purple-colored, hard berries, which are about 1–2cm long and 1cm wide and can resist temperatures below 40 °C [12]’

-Line 54: Please specify the health benefits.

à Line 54-55: Thank you for your advice, we modified the manuscript as ……’health benefits to prevent chronic diseases such as diabetes mellitus, cardiovascular diseases [13].’

-Lines 75-76: Please provide more detail on the study aims, in terms of the specific variables under investigation and units of measure.

à Line 76-79: Thank you for your thoughtful comments, we missed writing important information. According to your advice, we added the aim of this study like this ‘Based on these findings, the present study was designed to provide basic data for a better understanding of the pharmacological effects of HB extract (HBE) on the improvement of NAFLD using FFA-treated HepG2 cells and high fat diet (HFD) obese mouse.’

Methods:

-Lines 80-85: Please provide references to justify the methods used for the preparation of HBE.

à Line 82-88: According to your advice, we added references for the preparation of HBE.

-Lines 147-148: Please specify details on the quantification of food intake and body mass (e.g., the time of day for the measurement). Please replace 'body weight' with 'body mass'.

à Lines 164-165: We checked food intake as measure the remainder of what mouse eat every other day at 10 to 11 a.m., and body mass was measured weekly for 6 weeks. According to your advice, we replaced 'body weight' with 'body mass’.

Results:

-Line 181: Please replace the word 'substantial' with 'significant' if there was in fact no significant difference.

à Lines 225: Thank you for your insightful comments. According to your advice, we replace the word 'substantial' with 'significant'.

-Line 179: Please re-phrase this sentence so that it begins with, 'The CCK-8 assay indicated that...'

à Lines 223: According to your advice, we re-phrased sentence as ‘The CCK-8 assay indicated that cell viability on the effects of various concentrations of HBE plus 1mM FFA.’ 

-Lines 186-188, and lines 196-197: Please move this paragraph to the Methods.

à According to your advice, lines 186-188 and lines 196-197 were moved to Methods.

-Lines 235-238: Please move the text in these lines to the Discussion.

à According to your advice, lines 235-238 were moved to the Discussion.

-Lines 240-241, lines 261-262, lines 275-276, and lines 292-293: Please move the text in these lines to the Methods.

à According to your advice, lines 240-241, lines 275-276, and lines 292-293 were moved to the Methods.

-Lines 298-300: Please move this text to the Discussion.

à According to your advice, lines 298-300 was moved to the Discussion.

Discussion:

-Line 310: It is suggested that the Discussion begins with a statement of findings of the current study, rather than referring to findings of previous studies.

à Thank you very much for your consideration. overall overallAccording to your advice, begins of Discussion were a statement of our finding of this study.

-Line 341: Please provide a reference for this statement.

à Line 380-381: According to your advice, we added a reference for this statement.

Round  2

Reviewer 1 Report

The authors have responded to each and every one of the comments.
The work in its current form is accepted for publication.